# Deep Learning with Topological Signatures

**Christoph Hofer**
Department of Computer Science
University of Salzburg, Austria
chofer@cosy.sbg.ac.at

**Roland Kwitt**
Department of Computer Science
University of Salzburg, Austria
Roland.Kwitt@sbg.ac.at

**Marc Niethammer**
UNC Chapel Hill, NC, USA
mn@cs.unc.edu

**Andreas Uhl**
Department of Computer Science
University of Salzburg, Austria
uhl@cosy.sbg.ac.at

## Abstract

Inferring topological and geometrical information from data can offer an alternative perspective on machine learning problems. Methods from topological data analysis, e.g., persistent homology, enable us to obtain such information, typically in the form of summary representations of topological features. However, such topological signatures often come with an unusual structure (e.g., multisets of intervals) that is highly impractical for most machine learning techniques. While many strategies have been proposed to map these topological signatures into machine learning compatible representations, they suffer from being agnostic to the target learning task. In contrast, we propose a technique that enables us to input topological signatures to deep neural networks and *learn* a task-optimal representation during training. Our approach is realized as a novel input layer with favorable theoretical properties. Classification experiments on 2D object shapes and social network graphs demonstrate the versatility of the approach and, in case of the latter, we even outperform the state-of-the-art by a large margin.

## 1 Introduction

Methods from algebraic topology have only recently emerged in the machine learning community, most prominently under the term *topological data analysis (TDA)* [7]. Since TDA enables us to infer relevant topological and geometrical information from data, it can offer a novel and potentially beneficial perspective on various machine learning problems. Two compelling benefits of TDA are (1) its versatility, i.e., we are not restricted to any particular kind of data (such as images, sensor measurements, time-series, graphs, etc.) and (2) its robustness to noise. Several works have demonstrated that TDA can be beneficial in a diverse set of problems, such as studying the manifold of natural image patches [8], analyzing activity patterns of the visual cortex [28], classification of 3D surface meshes [27, 22], clustering [11], or recognition of 2D object shapes [29].

Currently, the most widely-used tool from TDA is *persistent homology* [15, 14]. Essentially[1], persistent homology allows us to track topological changes as we analyze data at multiple "scales". As the scale changes, topological features (such as connected components, holes, etc.) appear and disappear. Persistent homology associates a *lifespan* to these features in the form of a *birth* and a *death* time. The collection of (birth, death) tuples forms a multiset that can be visualized as a persistence diagram or a barcode, also referred to as a *topological signature* of the data. However, leveraging these signatures for learning purposes poses considerable challenges, mostly due to their

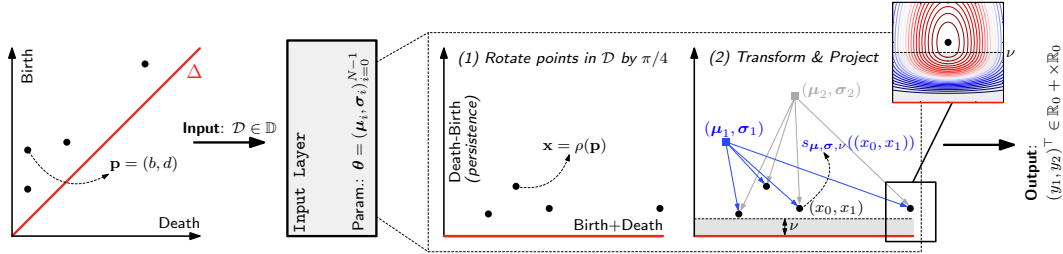

**Figure 1:** Illustration of the proposed network *input layer* for topological signatures. Each signature, in the form of a persistence diagram $\mathcal{D} \in \mathbb{D}$ (*left*), is projected w.r.t. a collection of *structure elements*. The layer's learnable parameters $\boldsymbol{\theta}$ are the locations $\boldsymbol{\mu}_i$ and the scales $\boldsymbol{\sigma}_i$ of these elements; $\nu \in \mathbb{R}^+$ is set a-priori and meant to discount the impact of points with low persistence (and, in many cases, of low discriminative power). The layer output $\mathbf{y}$ is a concatenation of the projections. In this illustration, $N = 2$ and hence $\mathbf{y} = (y_1, y_2)^\top$.

unusual structure as a multiset. While there exist suitable metrics to compare signatures (e.g., the Wasserstein metric), they are highly impractical for learning, as they require solving optimal matching problems.

**Related work.** In order to deal with these issues, several strategies have been proposed. In [2] for instance, Adcock et al. use invariant theory to "coordinatize" the space of barcodes. This allows to map barcodes to vectors of fixed size which can then be fed to standard machine learning techniques, such as support vector machines (SVMs). Alternatively, Adams et al. [1] map barcodes to so-called *persistence images* which, upon discretization, can also be interpreted as vectors and used with standard learning techniques. Along another line of research, Bubenik [6] proposes a mapping of barcodes into a Banach space. This has been shown to be particularly viable in a statistical context (see, e.g., [10]). The mapping outputs a representation referred to as a *persistence landscape*. Interestingly, under a specific choice of parameters, barcodes are mapped into $L_2(\mathbb{R}^2)$ and the inner-product in that space can be used to construct a valid kernel function. Similar, kernel-based techniques, have also recently been studied by Reininghaus et al. [27], Kwitt et al. [20] and Kusano et al. [19].

While all previously mentioned approaches retain certain stability properties of the original representation with respect to common metrics in TDA (such as the Wasserstein or Bottleneck distances), they also share one common *drawback*: the mapping of topological signatures to a representation that is compatible with existing learning techniques is *pre-defined*. Consequently, it is fixed and therefore *agnostic* to any specific learning task. This is clearly suboptimal, as the eminent success of deep neural networks (e.g., [18, 17]) has shown that *learning* representations is a preferable approach. Furthermore, techniques based on kernels [27, 20, 19] for instance, additionally suffer scalability issues, as training typically scales poorly with the number of samples (e.g., roughly cubic in case of kernel-SVMs). In the spirit of end-to-end training, we therefore aim for an approach that allows to learn a *task-optimal* representation of topological signatures. We additionally remark that, e.g., Qi et al. [25] or Ravanbakhsh et al. [26] have proposed architectures that can handle *sets*, but only with fixed size. In our context, this is impractical as the capability of handling sets with varying cardinality is a requirement to handle persistent homology in a machine learning setting.

**Contribution**. To realize this idea, we advocate a novel input layer for deep neural networks that takes a topological signature (in our case, a persistence diagram), and computes a parametrized projection that can be learned during network training. Specifically, this layer is designed such that its output is stable with respect to the 1-Wasserstein distance (similar to [27] or [1]). To demonstrate the versatility of this approach, we present experiments on 2D object shape classification and the classification of social network graphs. On the latter, we improve the state-of-the-art by a large margin, clearly demonstrating the power of combining TDA with deep learning in this context.

## 2 Background

For space reasons, we only provide a brief overview of the concepts that are relevant to this work and refer the reader to [16] or [14] for further details.

**Homology.** The key concept of homology theory is to study the properties of some object $X$ by means of (commutative) algebra. In particular, we assign to $X$ a sequence of modules $C_0, C_1, \ldots$

which are connected by homomorphisms $\partial_n : C_n \to C_{n-1}$ such that $\operatorname{im} \partial_{n+1} \subseteq \ker \partial_n$. A structure of this form is called a *chain complex* and by studying its homology groups $H_n = \ker \partial_n / \operatorname{im} \partial_{n+1}$ we can derive properties of $X$.

A prominent example of a homology theory is *simplicial homology*. Throughout this work, it is the used homology theory and hence we will now concretize the already presented ideas. Let $K$ be a simplicial complex and $K_n$ its $n$-skeleton. Then we set $C_n(K)$ as the vector space generated (freely) by $K_n$ over $\mathbb{Z}/2\mathbb{Z}$[2]. The connecting homomorphisms $\partial_n : C_n(K) \to C_{n-1}(K)$ are called boundary operators. For a simplex $\sigma = [x_0, \ldots, x_n] \in K_n$, we define them as $\partial_n(\sigma) = \sum_{i=0}^{n} [x_0, \ldots, x_{i-1}, x_{i+1}, \ldots, x_n]$ and linearly extend this to $C_n(K)$, i.e., $\partial_n(\sum \sigma_i) = \sum \partial_n(\sigma_i)$.

**Persistent homology.** Let $K$ be a simplicial complex and $(K^i)_{i=0}^m$ a sequence of simplicial complexes such that $\emptyset = K^0 \subseteq K^1 \subseteq \cdots \subseteq K^m = K$. Then, $(K^i)_{i=0}^m$ is called a *filtration* of $K$. If we use the extra information provided by the filtration of $K$, we obtain the following sequence of chain complexes (*left*),

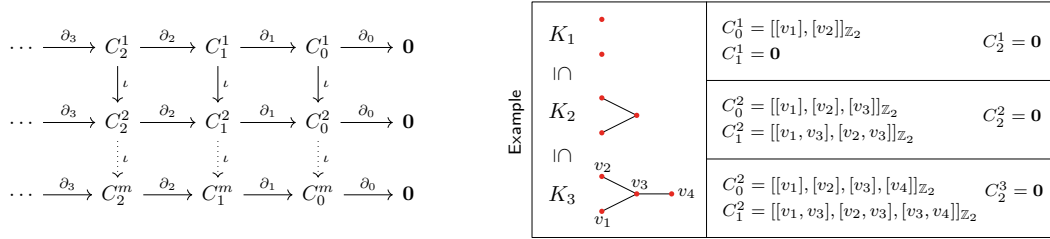

where $C_n^i = C_n(K_n^i)$ and $\iota$ denotes the inclusion. This then leads to the concept of *persistent homology groups*, defined by

$$H_n^{i,j} = \ker \partial_n^i / (\operatorname{im} \partial_{n+1}^j \cap \ker \partial_n^i) \quad \text{for} \quad i \le j \ .$$

The ranks, $\beta_n^{i,j} = \operatorname{rank} H_n^{i,j}$, of these homology groups (i.e., the *$n$-th persistent Betti numbers*), capture the number of homological features of dimensionality $n$ (e.g., connected components for $n = 0$, holes for $n = 1$, etc.) that persist from $i$ to (at least) $j$. In fact, according to [14, Fundamental Lemma of Persistent Homology], the quantities

$$\mu_n^{i,j} = (\beta_n^{i,j-1} - \beta_n^{i,j}) - (\beta_n^{i-1,j-1} - \beta_n^{i-1,j}) \quad \text{for} \quad i < j \tag{1}$$

encode all the information about the persistent Betti numbers of dimension $n$.

**Topological signatures**. A typical way to obtain a filtration of $K$ is to consider sublevel sets of a function $f : C_0(K) \to \mathbb{R}$. This function can be easily lifted to higher-dimensional chain groups of $K$ by

$$f([v_0, \ldots, v_n]) = \max\{f([v_i]) : 0 \le i \le n\} \ .$$

Given $m = |f(C_0(K))|$, we obtain $(K_i)_{i=0}^m$ by setting $K_0 = \emptyset$ and $K_i = f^{-1}((-\infty, a_i])$ for $1 \le i \le m$, where $a_1 < \cdots < a_m$ is the sorted sequence of values of $f(C_0(K))$. If we construct a multiset such that, for $i < j$, the point $(a_i, a_j)$ is inserted with multiplicity $\mu_n^{i,j}$, we effectively encode the persistent homology of dimension $n$ w.r.t. the sublevel set filtration induced by $f$. Upon adding diagonal points with infinite multiplicity, we obtain the following structure:

**Definition 1** (Persistence diagram). *Let* $\Delta = \{x \in \mathbb{R}_\Delta^2 : \operatorname{mult}(x) = \infty\}$ *be the multiset of the diagonal* $\mathbb{R}_\Delta^2 = \{(x_0, x_1) \in \mathbb{R}^2 : x_0 = x_1\}$, *where* $\operatorname{mult}$ *denotes the multiplicity function and let* $\mathbb{R}_\star^2 = \{(x_0, x_1) \in \mathbb{R}^2 : x_1 > x_0\}$. *A persistence diagram,* $\mathcal{D}$, *is a multiset of the form*

$$\mathcal{D} = \{x : x \in \mathbb{R}_\star^2\} \cup \Delta \ .$$

*We denote by* $\mathbb{D}$ *the set of all persistence diagrams of the form* $|\mathcal{D} \setminus \Delta| < \infty$ .

For a given complex $K$ of dimension $n_{\max}$ and a function $f$ (of the discussed form), we can interpret persistent homology as a mapping $(K, f) \mapsto (\mathcal{D}_0, \ldots, \mathcal{D}_{n_{\max}-1})$, where $\mathcal{D}_i$ is the diagram of dimension $i$ and $n_{\max}$ the dimension of $K$. We can additionally add a metric structure to the space of persistence diagrams by introducing the notion of distances.

**Definition 2** (Bottleneck, Wasserstein distance). *For two persistence diagrams $\mathcal{D}$ and $\mathcal{E}$, we define their Bottleneck ($\mathrm{w}_\infty$) and Wasserstein ($\mathrm{w}_p^q$) distances by*

$$\mathrm{w}_\infty(\mathcal{D},\mathcal{E}) = \inf_\eta \sup_{\mathbf{x}\in\mathcal{D}} ||\mathbf{x}-\eta(\mathbf{x})||_\infty \;\; and \;\; \mathrm{w}_p^q(\mathcal{D},\mathcal{E}) = \inf_\eta \left( \sum_{\mathbf{x}\in\mathcal{D}} ||\mathbf{x}-\eta(\mathbf{x})||_q^p \right)^{\frac{1}{p}}, \quad (2)$$

*where $p,q \in \mathbb{N}$ and the infimum is taken over all bijections $\eta : \mathcal{D} \to \mathcal{E}$.*

Essentially, this facilitates studying stability/continuity properties of topological signatures w.r.t. metrics in the filtration or complex space; we refer the reader to [12],[13], [9] for a selection of important stability results.

**Remark.** *By setting $\mu_n^{i,\infty} = \beta_n^{i,m} - \beta_n^{i-1,m}$, we extend Eq. (1) to features which never disappear, also referred to as essential. This change can be lifted to $\mathbb{D}$ by setting $\mathbb{R}_\star^2 = \{(x_0,x_1) \in \mathbb{R} \times (\mathbb{R} \cup \{\infty\}) : x_1 > x_0\}$. In Sec. 5, we will see that essential features can offer discriminative information.*

## 3    A network layer for topological signatures

In this section, we introduce the proposed (parametrized) network layer for topological signatures (in the form of persistence diagrams). The key idea is to take any $\mathcal{D}$ and define a projection w.r.t. a collection (of fixed size $N$) of *structure elements*.

In the following, we set $\mathbb{R}^+ := \{x \in \mathbb{R} : x > 0\}$ and $\mathbb{R}_0^+ := \{x \in \mathbb{R} : x \geq 0\}$, resp., and start by rotating points of $\mathcal{D}$ such that points on $\mathbb{R}_\Delta^2$ lie on the $x$-axis, see Fig. 1. The $y$-axis can then be interpreted as the *persistence* of features. Formally, we let $\mathbf{b}_0$ and $\mathbf{b}_1$ be the unit vectors in directions $(1,1)^\top$ and $(-1,1)^\top$ and define a mapping $\rho : \mathbb{R}_\star^2 \cup \mathbb{R}_\Delta^2 \to \mathbb{R} \times \mathbb{R}_0^+$ such that $\mathbf{x} \mapsto (\langle\mathbf{x},\mathbf{b}_0\rangle,\langle\mathbf{x},\mathbf{b}_1\rangle)$. This rotates points in $\mathbb{R}_\star \cup \mathbb{R}_\Delta^2$ clock-wise by $\pi/4$. We will later see that this construction is beneficial for a closer analysis of the layers' properties.

Similar to [27, 19], we choose exponential functions as structure elements, but other choices are possible (see Lemma 1). Differently to [27, 19], however, our structure elements *are not* at fixed locations (i.e., one element per point in $\mathcal{D}$), but their locations and scales are learned during training.

**Definition 3.** *Let $\boldsymbol{\mu} = (\mu_0,\mu_1)^\top \in \mathbb{R} \times \mathbb{R}^+, \boldsymbol{\sigma} = (\sigma_0,\sigma_1) \in \mathbb{R}^+ \times \mathbb{R}^+$ and $\nu \in \mathbb{R}^+$. We define*

$$s_{\boldsymbol{\mu},\boldsymbol{\sigma},\nu} : \mathbb{R} \times \mathbb{R}_0^+ \to \mathbb{R}$$

*as follows:*

$$s_{\boldsymbol{\mu},\boldsymbol{\sigma},\nu}\big((x_0,x_1)\big) = \begin{cases} e^{-\sigma_0^2(x_0-\mu_0)^2-\sigma_1^2(x_1-\mu_1)^2}, & x_1 \in [\nu,\infty) \\[2mm] e^{-\sigma_0^2(x_0-\mu_0)^2-\sigma_1^2(\ln(\frac{x_1}{\nu})+\nu-\mu_1)^2}, & x_1 \in (0,\nu) \\[2mm] 0, & x_1 = 0 \end{cases} \quad (3)$$

*A persistence diagram $\mathcal{D}$ is then projected w.r.t. $s_{\boldsymbol{\mu},\boldsymbol{\sigma},\nu}$ via*

$$S_{\boldsymbol{\mu},\boldsymbol{\sigma},\nu} : \mathbb{D} \to \mathbb{R}, \qquad \mathcal{D} \mapsto \sum_{\mathbf{x}\in\mathcal{D}} s_{\boldsymbol{\mu},\boldsymbol{\sigma},\nu}(\rho(\mathbf{x})) \;. \quad (4)$$

**Remark.** *Note that $s_{\boldsymbol{\mu},\boldsymbol{\sigma},\nu}$ is continuous in $x_1$ as*

$$\lim_{x\to\nu} x = \lim_{x\to\nu} \ln\left(\frac{x}{\nu}\right) + \nu \quad and \quad \lim_{x_1\to 0} s_{\boldsymbol{\mu},\boldsymbol{\sigma},\nu}\big((x_0,x_1)\big) = 0 = s_{\boldsymbol{\mu},\boldsymbol{\sigma},\nu}\big((x_0,0)\big)$$

*and $e^{(\cdot)}$ is continuous. Further, $s_{\boldsymbol{\mu},\boldsymbol{\sigma},\nu}$ is differentiable on $\mathbb{R} \times \mathbb{R}^+$, since*

$$1 = \lim_{x\to\nu^+} \frac{\partial x_1}{\partial x_1}(x) \quad and \quad \lim_{x\to\nu^-} \frac{\partial \left(\ln\left(\frac{x_1}{\nu}\right)+\nu\right)}{\partial x_1}(x) = \lim_{x\to\nu^-} \frac{\nu}{x} = 1 \;.$$

Also note that we use the log-transform in Eq. (4) to guarantee that $s_{\boldsymbol{\mu},\boldsymbol{\sigma},\nu}$ satisfies the conditions of Lemma 1; this is, however, *only one* possible choice.

**Remark.** *The intuition behind $\nu$ is the following. It is the threshold at which the log-transform starts to operate. The log-transform, on the other hand, stretches the space between the $x$-axis and the line drawn at $x + \nu$ to infinite length. As a consequence, $s_{\boldsymbol{\mu},\boldsymbol{\sigma},\nu} = 0$ for $x \in \mathbb{R}^2_\Delta$. This is necessary since otherwise $S_{\boldsymbol{\mu},\boldsymbol{\sigma},\nu}(\mathcal{D}) = \infty$ for $\mathcal{D} \in \mathbb{D}$ (as each persistence diagram contains points at the diagonal with infinite multiplicity).*

Finally, given a collection of $S_{\boldsymbol{\mu}_i,\boldsymbol{\sigma}_i,\nu}$, we combine them to form the output of the network layer.

**Definition 4.** *Let $N \in \mathbb{N}$, $\boldsymbol{\theta} = (\boldsymbol{\mu}_i, \boldsymbol{\sigma}_i)_{i=0}^{N-1} \in \left( (\mathbb{R} \times \mathbb{R}^+) \times (\mathbb{R}^+ \times \mathbb{R}^+) \right)^N$ and $\nu \in \mathbb{R}^+$. We define*

$$\mathcal{S}_{\boldsymbol{\theta},\nu} : \mathbb{D} \to (\mathbb{R}_0^+)^N \quad \mathcal{D} \mapsto \left( S_{\boldsymbol{\mu}_i,\boldsymbol{\sigma}_i,\nu}(\mathcal{D}) \right)_{i=0}^{N-1}.$$

*as the concatenation of all $N$ mappings defined in Eq. (4).*

Importantly, a network layer implementing Def. 4 is trainable via backpropagation, as (1) $s_{\boldsymbol{\mu}_i,\boldsymbol{\sigma}_i,\nu}$ is differentiable in $\boldsymbol{\mu}_i, \boldsymbol{\sigma}_i$, (2) $S_{\boldsymbol{\mu}_i,\boldsymbol{\sigma}_i,\nu}(\mathcal{D})$ is a finite sum of $s_{\boldsymbol{\mu}_i,\boldsymbol{\sigma}_i,\nu}$ and (3) $\mathcal{S}_{\boldsymbol{\theta},\nu}$ is just a concatenation.

# 4 Theoretical properties

In this section, we demonstrate that the proposed layer is stable w.r.t. the 1-Wasserstein distance $\mathrm{w}_1^q$, see Eq. (2). In fact, this claim will follow from a more general result, stating sufficient conditions on functions $s : \mathbb{R}^2_\star \cup \mathbb{R}^2_\Delta \to \mathbb{R}_0^+$ such that a construction in the form of Eq. (3) is stable w.r.t. $\mathrm{w}_1^q$.

**Lemma 1.** *Let*

$$s : \mathbb{R}^2_\star \cup \mathbb{R}^2_\Delta \to \mathbb{R}_0^+$$

*have the following properties:*

*(i) $s$ is Lipschitz continuous w.r.t. $\| \cdot \|_q$ and constant $K_s$*

*(ii) $s(\mathbf{x}) = 0$, for $\mathbf{x} \in \mathbb{R}^2_\Delta$*

*Then, for two persistence diagrams $\mathcal{D}, \mathcal{E} \in \mathbb{D}$, it holds that*

$$\left| \sum_{x \in \mathcal{D}} s(x) - \sum_{y \in \mathcal{E}} s(y) \right| \leq K_s \cdot \mathrm{w}_1^q(\mathcal{D}, \mathcal{E}) \ . \tag{5}$$

*Proof.* see Appendix B

**Remark.** *At this point, we want to clarify that Lemma 1 is not specific to $s_{\boldsymbol{\mu},\boldsymbol{\sigma},\nu}$ (e.g., as in Def. 3). Rather, Lemma 1 yields sufficient conditions to construct a $\mathrm{w}_1$-stable input layer. Our choice of $s_{\boldsymbol{\mu},\boldsymbol{\sigma},\nu}$ is just a natural example that fulfils those requirements and, hence, $\mathcal{S}_{\boldsymbol{\theta},\nu}$ is just one possible representative of a whole family of input layers.*

With the result of Lemma 1 in mind, we turn to the specific case of $\mathcal{S}_{\boldsymbol{\theta},\nu}$ and analyze its stability properties w.r.t. $\mathrm{w}_1^q$. The following lemma is important in this context.

**Lemma 2.** *$s_{\boldsymbol{\mu},\boldsymbol{\sigma},\nu}$ has absolutely bounded first-order partial derivatives w.r.t. $x_0$ and $x_1$ on $\mathbb{R} \times \mathbb{R}^+$.*

*Proof.* see Appendix B

**Theorem 1.** *$\mathcal{S}_{\boldsymbol{\theta},\nu}$ is Lipschitz continuous with respect to $\mathrm{w}_1^q$ on $\mathbb{D}$.*

*Proof.* Lemma 2 immediately implies that $s_{\boldsymbol{\mu},\boldsymbol{\sigma},\nu}$ from Eq. (3) is Lipschitz continuous w.r.t $|| \cdot ||_q$. Consequently, $s = s_{\boldsymbol{\mu},\boldsymbol{\sigma},\nu} \circ \rho$ satisfies property (i) from Lemma 1; property (ii) from Lemma 1 is satisfied by construction. Hence, $S_{\boldsymbol{\mu},\boldsymbol{\sigma},\nu}$ is Lipschitz continuous w.r.t. $\mathrm{w}_1^q$. Consequently, $\mathcal{S}_{\boldsymbol{\theta},\nu}$ is Lipschitz in each coordinate and therefore Liptschitz continuous. □

Interestingly, the stability result of Theorem 1 is comparable to the stability results in [1] or [27] (which are also w.r.t. $\mathrm{w}_1^q$ and in the setting of diagrams with finitely-many points). However, contrary to previous works, if we would chop-off the input layer after network training, we would then have a mapping $\mathcal{S}_{\boldsymbol{\theta},\nu}$ of persistence diagrams that is *specifically-tailored to the learning task* on which the network was trained.

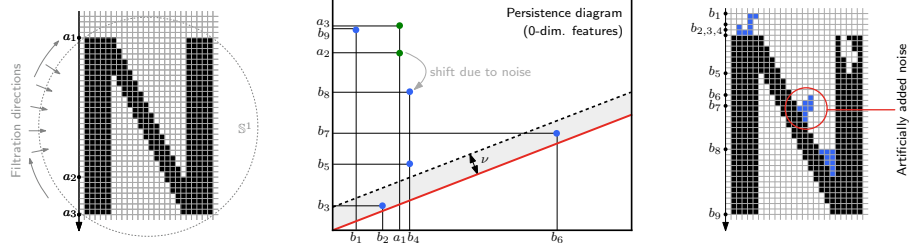

**Figure 2:** Height function filtration of a "clean" (*left*, green points) and a "noisy" (*right*, blue points) shape along direction $\mathbf{d} = (0, -1)^\top$. This example demonstrates the insensitivity of homology towards noise, as the added noise only (1) slightly shifts the dominant points (upper left corner) and (2) produces additional points close to the diagonal, which have little impact on the Wasserstein distance and the output of our layer.

## 5 Experiments

To demonstrate the versatility of the proposed approach, we present experiments with two totally different types of data: (1) 2D shapes of objects, represented as binary images and (2) social network graphs, given by their adjacency matrix. In both cases, the learning task is *classification*. In each experiment we ensured a balanced group size (per label) and used a 90/10 random training/test split; all reported results are averaged over five runs with fixed $\nu = 0.1$. In practice, points in input diagrams were thresholded at $0.01$ for computational reasons. Additionally, we conducted a reference experiment on all datasets using simple vectorization (see Sec. 5.3) of the persistence diagrams in combination with a linear SVM.

**Implementation**. All experiments were implemented in `PyTorch`[3], using `DIPHA`[4] and `Perseus` [23]. Source code is publicly-available at `https://github.com/c-hofer/nips2017`.

### 5.1 Classification of 2D object shapes

We apply persistent homology combined with our proposed input layer to two different datasets of binary 2D object shapes: (1) the `Animal` dataset, introduced in [3] which consists of 20 different animal classes, 100 samples each; (2) the `MPEG-7` dataset which consists of 70 classes of different object/animal contours, 20 samples each (see [21] for more details).

**Filtration.** The requirements to use persistent homology on 2D shapes are twofold: *First*, we need to assign a simplicial complex to each shape; *second*, we need to appropriately filtrate the complex. While, in principle, we could analyze contour features, such as curvature, and choose a sublevel set filtration based on that, such a strategy requires substantial preprocessing of the discrete data (e.g., smoothing). Instead, we choose to work with the raw pixel data and leverage the *persistent homology transform*, introduced by Turner et al. [29]. The filtration in that case is based on sublevel sets of the *height function*, computed from multiple directions (see Fig. 2). Practically, this means that we *directly construct a simplicial complex from the binary image*. We set $K_0$ as the set of all pixels which are contained in the object. Then, a 1-simplex $[\mathbf{p}_0, \mathbf{p}_1]$ is in the 1-skeleton $K_1$ iff $\mathbf{p}_0$ and $\mathbf{p}_1$ are 4–neighbors on the pixel grid. To filtrate the constructed complex, we denote by $\mathbf{b}$ the barycenter of the object and with $r$ the radius of its bounding circle around $\mathbf{b}$. Finally, we define, for $[\mathbf{p}] \in K_0$ and $\mathbf{d} \in \mathbb{S}^1$, the filtration function by $f([p]) = 1/r \cdot \langle \mathbf{p} - \mathbf{b}, \mathbf{d} \rangle$. Function values are lifted to $K_1$ by taking the maximum, cf. Sec. 2. Finally, let $\mathbf{d}_i$ be the 32 equidistantly distributed directions in $\mathbb{S}^1$, starting from $(1, 0)^\top$. For each shape, we get a vector of persistence diagrams $(\mathcal{D}_i)_{i=1}^{32}$ where $\mathcal{D}_i$ is the 0-th diagram obtained by filtration along $\mathbf{d}_i$. As most objects do not differ in homology groups of *higher* dimensions ($> 0$), we did not use the corresponding persistence diagrams.

**Network architecture.** While the full network is listed in the *supplementary material*, the key architectural choices are: 32 independent input branches, i.e., one for each filtration direction. Further, the $i$-th branch gets, as input, the vector of persistence diagrams from directions $\mathbf{d}_{i-1}, \mathbf{d}_i$ and $\mathbf{d}_{i+1}$. This is a straightforward approach to capture dependencies among the filtration directions. We use cross-entropy loss to train the network for $400$ epochs, using stochastic gradient descent (SGD) with mini-batches of size $128$ and an initial learning rate of $0.1$ (halved every 25-th epoch).

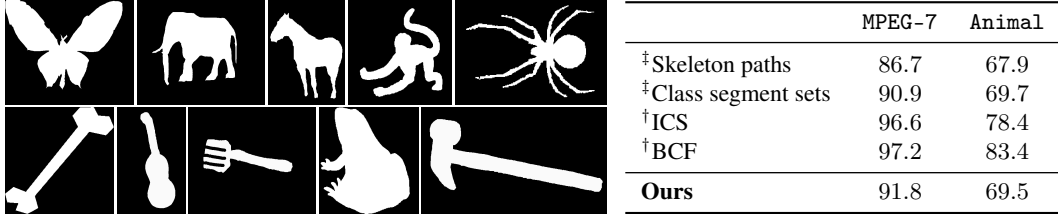

|  | MPEG-7 | Animal |
|---|---|---|
| ‡Skeleton paths | 86.7 | 67.9 |
| ‡Class segment sets | 90.9 | 69.7 |
| †ICS | 96.6 | 78.4 |
| †BCF | 97.2 | 83.4 |
| **Ours** | 91.8 | 69.5 |

**Figure 3:** *Left*: some examples from the MPEG-7 (*bottom*) and Animal (*top*) datasets. *Right*: Classification results, compared to the two best (†) and two worst (‡) results reported in [30].

**Results.** Fig. 3 shows a selection of 2D object shapes from both datasets, together with the obtained classification results. We list the two best (†) and two worst (‡) results as reported in [30]. While, on the one hand, using topological signatures is below the state-of-the-art, the proposed architecture is still better than other approaches that are specifically tailored to the problem. Most notably, our approach *does not* require any specific data preprocessing, whereas all other competitors listed in Fig. 3 require, e.g., some sort of contour extraction. Furthermore, the proposed architecture readily generalizes to 3D with the only difference that in this case $\mathbf{d}_i \in \mathbb{S}^2$. Fig. 4 (*Right*) shows an exemplary visualization of the position of the learned structure elements for the Animal dataset.

## 5.2 Classification of social network graphs

In this experiment, we consider the problem of graph classification, where vertices are unlabeled and edges are undirected. That is, a graph $\mathcal{G}$ is given by $\mathcal{G} = (V, E)$, where $V$ denotes the set of vertices and $E$ denotes the set of edges. We evaluate our approach on the challenging problem of social network classification, using the two largest benchmark datasets from [31], i.e., reddit-5k (5 classes, 5k graphs) and reddit-12k (11 classes, ≈12k graphs). Each sample in these datasets represents a discussion graph and the classes indicate *subreddits* (e.g., *worldnews*, *video*, etc.).

**Filtration.** The construction of a simplicial complex from $\mathcal{G} = (V, E)$ is straightforward: we set $K_0 = \{[v] \in V\}$ and $K_1 = \{[v_0, v_1] : \{v_0, v_1\} \in E\}$. We choose a very simple filtration based on the *vertex degree*, i.e., the number of incident edges to a vertex $v \in V$. Hence, for $[v_0] \in K_0$ we get $f([v_0]) = \deg(v_0)/\max_{v \in V} \deg(v)$ and again lift $f$ to $K_1$ by taking the maximum. Note that chain groups are trivial for dimension $> 1$, hence, all features in dimension 1 are *essential*.

**Network architecture.** Our network has four input branches: two for each dimension (0 and 1) of the homological features, split into *essential* and *non-essential* ones, see Sec. 2. We train the network for 500 epochs using SGD and cross-entropy loss with an initial learning rate of 0.1 (reddit_5k), or 0.4 (reddit_12k). The full network architecture is listed in the *supplementary material*.

**Results.** Fig. 5 (*right*) compares our proposed strategy to state-of-the-art approaches from the literature. In particular, we compare against (1) the graphlet kernel (GK) and deep graphlet kernel (DGK) results from [31], (2) the Patchy-SAN (PSCN) results from [24] and (3) a recently reported graph-feature + random forest approach (RF) from [4]. As we can see, using topological signatures in our proposed setting considerably outperforms the current state-of-the-art on both datasets. This is an interesting observation, as PSCN [24] for instance, also relies on node degrees and an extension of the convolution operation to graphs. Further, the results reveal that including *essential* features is key to these improvements.

## 5.3 Vectorization of persistence diagrams

Here, we briefly present a reference experiment we conducted following Bendich et al. [5]. The idea is to directly use the persistence diagrams as features via *vectorization*. For each point $(b, d)$ in a persistence diagram $\mathcal{D}$ we calculate its *persistence*, i.e., $d - b$. We then sort the calculated persistences by magnitude from high to low and take the first $N$ values. Hence, we get, for each persistence diagram, a vector of dimension $N$ (if $|\mathcal{D} \setminus \Delta| < N$, we pad with zero). We used this technique on all four data sets. As can be seen from the results in Table 4 (averaged over 10 cross-validation runs), vectorization performs poorly on MPEG-7 and Animal but can lead to competitive rates on reddit-5k and reddit-12k. Nevertheless, the obtained performance is considerably inferior to our proposed approach.

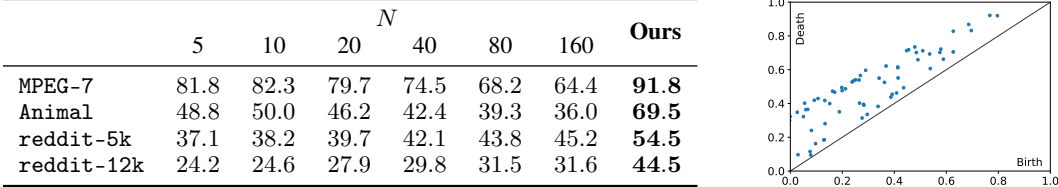

| | | | $N$ | | | | **Ours** |
|---|---|---|---|---|---|---|---|
| | 5 | 10 | 20 | 40 | 80 | 160 | |
| MPEG-7 | 81.8 | 82.3 | 79.7 | 74.5 | 68.2 | 64.4 | **91.8** |
| Animal | 48.8 | 50.0 | 46.2 | 42.4 | 39.3 | 36.0 | **69.5** |
| reddit-5k | 37.1 | 38.2 | 39.7 | 42.1 | 43.8 | 45.2 | **54.5** |
| reddit-12k | 24.2 | 24.6 | 27.9 | 29.8 | 31.5 | 31.6 | **44.5** |

**Figure 4:** *Left*: Classification accuracies for a linear SVM trained on vectorized (in $\mathbb{R}^N$) persistence diagrams (see Sec. 5.3). *Right*: Exemplary visualization of the learned structure elements (in 0-th dimension) for the `Animal` dataset and filtration direction $\mathbf{d} = (-1, 0)^\top$. Centers of the learned elements are marked in blue.

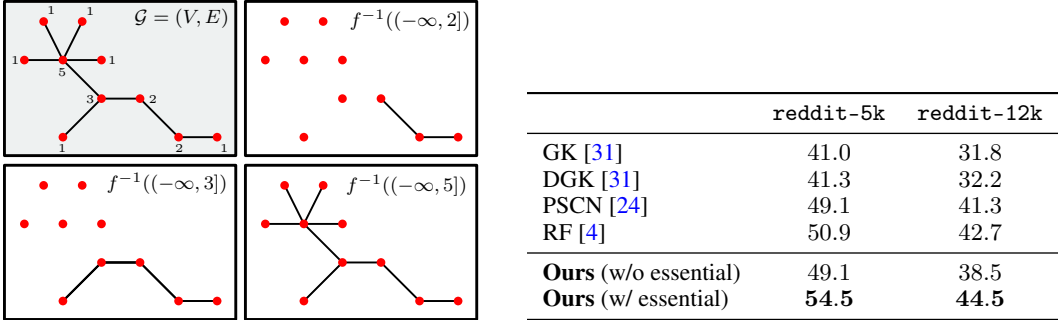

| | reddit-5k | reddit-12k |
|---|---|---|
| GK [31] | 41.0 | 31.8 |
| DGK [31] | 41.3 | 32.2 |
| PSCN [24] | 49.1 | 41.3 |
| RF [4] | 50.9 | 42.7 |
| **Ours** (w/o essential) | 49.1 | 38.5 |
| **Ours** (w/ essential) | **54.5** | **44.5** |

**Figure 5:** *Left*: Illustration of graph filtration by vertex degree, i.e., $f \equiv \deg$ (for different choices of $a_i$, see Sec. 2). *Right*: Classification results as reported in [31] for GK and DGK, Patchy-SAN (PSCN) as reported in [24] and feature-based random-forest (RF) classification from [4].

Finally, we remark that in both experiments, tests with the kernel of [27] turned out to be computationally impractical, (1) on shape data due to the need to evaluate the kernel for all filtration directions and (2) on graphs due the large number of samples and the number of points in each diagram.

## 6 Discussion

We have presented, to the best of our knowledge, the first approach towards learning *task-optimal* stable representations of topological signatures, in our case persistence diagrams. Our particular realization of this idea, i.e., as an input layer to deep neural networks, not only enables us to learn with topological signatures, but also to use them as additional (and potentially complementary) inputs to existing deep architectures. From a theoretical point of view, we remark that the presented *structure elements* are not restricted to exponential functions, so long as the conditions of Lemma 1 are met. One drawback of the proposed approach, however, is the artificial bending of the persistence axis (see Fig. 1) by a logarithmic transformation; in fact, other strategies might be possible and better suited in certain situations. A detailed investigation of this issue is left for future work. From a practical perspective, it is also worth pointing out that, in principle, the proposed layer could be used to handle any kind of input that comes in the form of multisets (of $\mathbb{R}^n$), whereas previous works only allow to handle sets of fixed size (see Sec. 1). In summary, we argue that our experiments show strong evidence that topological features of data can be beneficial in many learning tasks, not necessarily to replace existing inputs, but rather as a complementary source of discriminative information.

**Acknowledgements.** This work was partially funded by the Austrian Science Fund FWF (KLI project 00012) and the Spinal Cord Injury and Tissue Regeneration Center Salzburg (SCI-TReCS), Paracelsus Medical University, Salzburg.

# A   Technical results

**Lemma 3.** *Let $\alpha \in \mathbb{R}^+$ and $\beta \in \mathbb{R}$. We have*

*i)* $\quad \lim\limits_{x \to 0} \frac{\ln(x)}{x} \cdot e^{-\alpha(\ln(x)+\beta)^2} = 0$
$\qquad\qquad\qquad$
*ii)* $\quad \lim\limits_{x \to 0} \frac{1}{x} \cdot e^{-\alpha(\ln(x)+\beta)^2} = 0$ .

*Proof.* We omit the proof for brevity (see *supplementary material* for details), but remark that only i) needs to be shown as ii) follows immediately.

# B   Proofs

*Proof of Lemma 1.* Let $\varphi$ be a bijection between $\mathcal{D}$ and $\mathcal{E}$ which realizes $\mathrm{w}_1^q(\mathcal{D}, \mathcal{E})$ and let $\mathcal{D}_0 = \mathcal{D} \setminus \Delta$, $\mathcal{E}_0 = \mathcal{E} \setminus \Delta$. To show the result of Eq. (5), we consider the following decomposition:

$$
\begin{aligned}
\mathcal{D} &= \varphi^{-1}(\mathcal{E}_0) \cup \varphi^{-1}(\Delta) \\
&= \underbrace{(\varphi^{-1}(\mathcal{E}_0) \setminus \Delta)}_{A} \cup \underbrace{(\varphi^{-1}(\mathcal{E}_0) \cap \Delta)}_{B} \cup \underbrace{(\varphi^{-1}(\Delta) \setminus \Delta)}_{C} \cup \underbrace{(\varphi^{-1}(\Delta) \cap \Delta)}_{D}
\end{aligned}
\tag{6}
$$

Except for the term $D$, all sets are finite. In fact, $\varphi$ realizes the Wasserstein distance $\mathrm{w}_1^q$ which implies $\varphi\big|_D = \mathrm{id}$. Therefore, $s(x) = s(\varphi(x)) = 0$ for $x \in D$ since $D \subset \Delta$. Consequently, we can ignore $D$ in the summation and it suffices to consider $E = A \cup B \cup C$. It follows that

$$
\begin{aligned}
\left| \sum_{x \in \mathcal{D}} s(x) - \sum_{y \in \mathcal{E}} s(y) \right| &= \left| \sum_{x \in \mathcal{D}} s(x) - \sum_{x \in \mathcal{D}} s(\varphi(x)) \right| = \left| \sum_{x \in E} s(x) - \sum_{x \in E} s(\varphi(x)) \right| \\
&= \left| \sum_{x \in E} s(x) - s(\varphi(x)) \right| \leq \sum_{x \in E} |s(x) - s(\varphi(x))| \\
&\leq K_s \cdot \sum_{x \in E} ||x - \varphi(x)||_q = K_s \cdot \sum_{x \in \mathcal{D}} ||x - \varphi(x)||_q = K_s \cdot \mathrm{w}_1^q(\mathcal{D}, \mathcal{E}) \ .
\end{aligned}
$$

$\square$

*Proof of Lemma 2.* Since $s_{\boldsymbol{\mu},\boldsymbol{\sigma},\nu}$ is defined differently for $x_1 \in [\nu, \infty)$ and $x_1 \in (0, \nu)$, we need to distinguish these two cases. In the following $x_0 \in \mathbb{R}$.

(1) $x_1 \in [\nu, \infty)$: The partial derivative w.r.t. $x_i$ is given as

$$
\begin{aligned}
\left( \frac{\partial}{\partial x_i} s_{\boldsymbol{\mu},\boldsymbol{\sigma},\nu} \right)(x_0, x_1) &= C \cdot \left( \frac{\partial}{\partial x_i} e^{-\sigma_i^2(x_i - \mu_i)^2} \right)(x_0, x_1) \\
&= C \cdot e^{-\sigma_i^2(x_i - \mu_i)^2} \cdot (-2\sigma_i^2)(x_i - \mu_i) \ ,
\end{aligned}
\tag{7}
$$

where $C$ is just the part of $\exp(\cdot)$ which is not dependent on $x_i$. For all cases, i.e., $x_0 \to \infty, x_0 \to -\infty$ and $x_1 \to \infty$, it holds that Eq. (7) $\to 0$.

(2) $x_1 \in (0, \nu)$: The partial derivative w.r.t. $x_0$ is similar to Eq. (7) with the same asymptotic behaviour for $x_0 \to \infty$ and $x_0 \to -\infty$. However, for the partial derivative w.r.t. $x_1$ we get

$$
\begin{aligned}
\left( \frac{\partial}{\partial x_1} s_{\mu,\sigma,\nu} \right)(x_0, x_1) &= C \cdot \left( \frac{\partial}{\partial x_1} e^{-\sigma_1^2(\ln(\frac{x_1}{\nu})+\nu-\mu_1)^2} \right)(x_0, x_1) \\
&= C \cdot e^{(\cdots)} \cdot (-2\sigma_1^2) \cdot \left( \ln\left(\frac{x_1}{\nu}\right) + \nu - \mu_1 \right) \cdot \frac{\nu}{x_1} \\
&= C' \cdot \left( \underbrace{e^{(\cdots)} \cdot \left( \ln\left(\frac{x_1}{\nu}\right) \cdot \frac{1}{x_1} \right)}_{(a)} + (\nu - \mu_1) \cdot \underbrace{e^{(\cdots)} \cdot \frac{1}{x_1}}_{(b)} \right) \ .
\end{aligned}
\tag{8}
$$

As $x_1 \to 0$, we can invoke Lemma 3 i) to handle (a) and Lemma 3 ii) to handle (b); conclusively, Eq. (8) $\to 0$. As the partial derivatives w.r.t. $x_i$ are continuous and their limits are 0 on $\mathbb{R}$, $\mathbb{R}^+$, resp., we conclude that they are *absolutely bounded*. $\square$

## Footnotes

[1]We will make these concepts more concrete in Sec. 2.

[2]Simplicial homology is not specific to $\mathbb{Z}/2\mathbb{Z}$, but it's a typical choice, since it allows us to interpret $n$-chains as sets of $n$-simplices.

[3] `https://github.com/pytorch/pytorch`

[4] `https://bitbucket.org/dipha/dipha`

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
