[Supplementary Material]

# Deep Learning with Topological Signatures
## Supplementary Material

C. Hofer          R. Kwitt          M. Niethammer          A. Uhl

This supplementary material contains technical details that were left-out in the original submission for brevity. When necessary, we refer to the submitted manuscript by [Manuscript, Sec. XXX].

## 1 Additional proofs

In the manuscript, we omitted the proof for the following technical lemma. For completeness, the lemma is repeated and its proof is given below.

**Lemma 1.** *Let $\alpha \in \mathbb{R}^+$ and $\beta \in \mathbb{R}$. We have*

*(i)* $\lim_{x \to 0} \frac{\ln(x)}{x} \cdot e^{-\alpha(\ln(x)+\beta)^2} = 0$

*(ii)* $\lim_{x \to 0} \frac{1}{x} \cdot e^{-\alpha(\ln(x)+\beta)^2} = 0$ .

*Proof.* We only need to prove the first statement, as the second follows immediately. Hence, consider

$$
\begin{aligned}
\lim_{x \to 0} \frac{\ln(x)}{x} \cdot e^{-\alpha(\ln(x)+\beta)^2} &= \lim_{x \to 0} \ln(x) \cdot e^{-\ln(x)} \cdot e^{-\alpha(\ln(x)+\beta)^2} \\
&= \lim_{x \to 0} \ln(x) \cdot e^{-\alpha(\ln(x)+\beta)^2 - \ln(x)} \\
&= \lim_{x \to 0} \ln(x) \cdot \left( e^{\alpha(\ln(x)+\beta)^2 + \ln(x)} \right)^{-1} \\
&\overset{(*)}{=} \lim_{x \to 0} \frac{\partial}{\partial x} \ln(x) \cdot \left( \frac{\partial}{\partial x} e^{\alpha(\ln(x)+\beta)^2 + \ln(x)} \right)^{-1} \\
&= \lim_{x \to 0} \frac{1}{x} \cdot \left( e^{\alpha(\ln(x)+\beta)^2 + \ln(x)} \cdot \left( 2\alpha(\ln(x)+\beta)\frac{1}{x} + \frac{1}{x} \right) \right)^{-1} \\
&= \lim_{x \to 0} \left( e^{\alpha(\ln(x)+\beta)^2 + \ln(x)} \cdot (2\alpha(\ln(x)+\beta) + 1) \right)^{-1} \\
&= 0
\end{aligned}
$$

where we use de l'Hôpital's rule in $(*)$. $\qquad \square$

## 2 Network architectures

**2D object shape classification.** Fig. 1 illustrates the network architecture used for *2D object shape classification* in [Manuscript, Sec. 5.1]. Note that the persistence diagrams from three consecutive filtration directions $\mathbf{d}_i$ share one input layer. As we use 32 directions, we have 32 input branches. The convolution operation operates with kernels of size $1 \times 1 \times 3$ and a stride of 1. The max-pooling operates along the filter dimension. For better readability, we have added the output size of certain layers. We train with the network with stochastic gradient descent (SGD) and a mini-batch size of 128 for 300 epochs. Every 20th epoch, the learning rate (initially set to 0.1) is halved.

**Figure 1:** 2D object shape classification network architecture.

**Graph classification.** Fig. 2 illustrates the network architecture used for *graph classification* in [Manuscript, Sec. 5.2]. In detail, we have 3 input branches: first, we split 0-dimensional features into *essential* and *non-essential* ones; second, since there are only *essential* features in dimension 1, see [Manuscript, Sec. 5.2, **Filtration**], we do not need a branch for *non-essential* features. We train the network using SGD with mini-batches of size 128 for 300 epochs. The initial learning rate is set to 0.1 (reddit_5k) and 0.4 (reddit_12k), resp., and halved every 20th epochs.

**Figure 2:** Graph classification network architecture.

## 2.1 Technical handling of essential features

In case of of 2D object shapes, the death times of essential features are mapped to the max. filtration value and kept in the original persistence diagrams. In fact, for `Animal` and `MPEG-7`, there is always only one connected component and consequently only one *essential* feature in dimension 0 (i.e., it does not make sense to handle this one point in a separate input branch).

In case of social network graphs, essential features are mapped to the real line (using their birth time) and handled in separate input branches (see Fig. 2) with 1D structure elements. This is in contrast to the 2D object shape experiments, as we might have many essential features (in dimensions 0 *and* 1) that require handling in separate input branches.