[Reviews · NeurIPS 2017]

Reviewer 1



This paper proposes a deep neural network model to learn from persistence diagrams extracted from data. A persistence diagram is a 2D point sets describing the topological information of a given data in the view of a chosen scalar function. While a diagram describes useful global information of the data, existing learning methods [25,18] only use it in a kernel-based setting. The key contribution of this paper is to construct an input layer for persistence diagrams. This is non-trivial as persistence diagrams behave very differently from traditional vectorized features (non-Euclidean metric). The key idea is to learn a set of single-point diagrams, as well as parametrized distances from these diagrams to any training/testing diagram. This way, any input persistence diagram can be transformed into a set of distances from these single-point diagrams. And be passed to the next layer of the deep neural network. The locations of the single point (\mu) and the parameters for distance (\sigma) can be learned through the training accordingly. Two real world datasets are used to demonstrate the proposed method. On a 2D shape dataset, the proposed method performs on par with state-of-the-arts. On the network dataset, it outperforms state-of-the-arts significantly. I like the idea. The construction of the input layer from persistence diagrams is novel and well-thought. Experimental results are convincing. I did not check the proofs carefully. But they are believable to me. The presentation has a lot of room to improve. Notably the notations and explanations in page 4 could have been simplified and supplemented with more pictorial illustrations. First, the rotation of the diagrams by \pi/4 does not seem necessary to me. I am guessing the goal is to make x_1 the persistence so the learning result is more interpretable. But this was not discussed in the rest of the paper. Second, the presentation can be simplified by separating the diagram and the essential dots (so the first line of Eq. (3), and the notation R_\star can be removed). In fact, in experiment 1, essential dots are not used (as the shapes are generally trivial in topology). In experiment 2, they are separated from the main diagrams in the model. A unified presentation can be moved to the supplemental material completely to improve the readability of the paper. Third, R_\Delta, R_\Star, etc, should be replaced by other symbols. They look like extended Euclidean planes rather than point sets. Forth, the presentation could benefit from more figures and explanations. (The right half of Fig 1 is doing so, but with insufficient explanations.) - It would be much helpful to see some illustrations of the learning results on the real datasets. E.g., what are the single-point diagrams learned in these data. - I understand [25] does not work due to speed issues. How about simply vectorize the diagrams and use them as features? This will strengthen the result but demonstrating the necessity of deep neural networks?

Reviewer 2



The authors propose a “input layer” for a deep network that takes topological persistence diagrams as input and transforms them to representations that can be optimized by the network through back-propagation. The proposed transformations are proved to be Lipschitz continuous and differentiable. The merits of the proposed architecture are demonstrated using 2D shape classification, and graph classification experiments. Comments: 1. The authors have done a very good job in introducing persistent homology and persistence diagrams. The presentation of the paper is mostly smooth and easy to read. 2. The representation is well-thought, and is essentially a sum of likelihoods of the points in the PD with respect to 2D Gaussians whose mean and covariances are learned. 3. It is clear that (3) is continuous, but the remark below definition (3) should be clarified. For example, the expressions with the limits ignore the exponential term. 4. In order to prove Lemma 2, we need to show that the partial derivatives are bounded on the entire domain. However the proof only shows the bounds at the extreme values. Although it is easy to see, it may be worth mentioning how the derivatives are bounded in the interior of the domain. 5. What is the significance of Lemma 1? Does it connect to the stability of the representations with respect to the perturbations of function f? 6. The results of the 2D shape classification experiment are not very convincing although the methodology seems solid. The authors should explore more suitable applications such as 3D shape recognition (see [20]) where PDs have been shown to be a good representation. There are many applications where PD features are a good choice. Section 5.2’s results are good. Experiments seem like an area of the paper that can be improved (given that the paper proposes a deep learning system). Update: -------- The authors have answered most of my comments satisfactorily.

Reviewer 3



SUMMARY * This paper proposes to map topological signatures (eg multisets of intervals from persistent homology) to task dependent representations. * Whereas previous approaches have used predefined maps to extract features from the topological signatures, here it is proposed to learn this map. Training the representation appears to be a natural and important step. * The paper includes experiments demonstrating that the proposed approach can help making TDA methods more useful in classification problems. CLARITY * In think that the paper would benefit from a more intuitive and self contained background section. Maybe consider expanding the example to explain more of the abstract notions discussed in this section. RELEVANCE * Mapping topological signatures to useful representations is an important problem in TDA. This paper proposes an approach to using trainable features, in contrast to fixed features that have been considered in other works. * The new theoretical results are limited to proving a certain stability wrt the Wasserstein metric, a result which seems to be quite direct from the definitions. * One of the conclusions is that, while using topological signatures does not always reach state of the art in classification tasks, the proposed approach is better than previous approaches using topological signatures. MINOR * Fig 1 left, should birth not come before death? * In Def 1, what is the multiplicity function and why is \Delta defined to have an infinite value? * What is \mathcal{D}_i and n_max in line 104? * Explain or replace braket notation in line 122. * What is the role of \nu in (3)? CONCLUSION * In general I think that this is a valuable contribution towards increasing the usefulness of traditional TDA methods in the context of machine learning.